# Topology Optimization and Multiobjective Optimization for Drive Axle Housing of a Rear Axle Drive Truck

**DOI:** 10.3390/ma15155268

**Published:** 2022-07-30

**Authors:** Bin Zheng, Shengyan Fu, Jilin Lei

**Affiliations:** 1Yunnan Province Key Laboratory of Internal Combustion Engines, Kunming University of Science and Technology, Kunming 650500, China; 13438523134@163.com; 2School of Intelligent Manufacturing, Panzhihua University, Panzhihua 617000, China; 18111727154@163.com

**Keywords:** drive axle housing, finite element analysis, dynamic characteristics, static characteristics, fatigue analysis, topological optimization, response surface methodology, multiobjective optimization

## Abstract

As one of the important load-bearing components of a truck, the drive axle housing must meet the requirements of stiffness and strength. The traditional design method uses redundancy design to meet the performance requirements. The joint design between the three-dimensional mathematical model and finite element model is adopted, and the optimal design of the drive axle housing is realized based on topology optimization and multiobjective optimization. Firstly, the static analysis of the drive axle housing of a rear axle drive truck was carried out with four typical working conditions. It was concluded that the four working conditions all operate under the yield limit of the material, and it was found that the maximum equivalent stress of the four working conditions occurs at the step of the half-shaft casing. Among the four working conditions, the most critical one is the maximum vertical force working condition. Then, based on the maximum vertical force working condition, the fatigue life analysis is conducted, and the minimum fatigue life appears at the transition position of the half-shaft sleeve and the arc transition position of the main reducer chamber. The remaining parts can meet the design requirements. The overall safety factor of the drive axle housing is mainly between 1 and 5 when operating under this working condition. Then, through modal analysis, the first to sixth natural frequency and vibration modes of the drive axle housing are extracted. Based on the modal analysis, the dynamic characteristics of the drive axle housing are further studied by harmonic response analysis and random vibration analysis. Finally, two kinds of lightweight optimization schemes for the drive axle housing are given. Topology optimization reduces the mass of the drive axle housing by 17.4%, but the overall performance slightly decreases. Then, the five dimensional parameters of the drive axle housing are selected as design variables. The mass, maximum deformation, equivalent stress, service life, and the first-, second- and third-order natural frequencies are defined as objective functions. Through the optimal space-filling design method, the experimental designs are performed and the sample points are obtained. Based on the results of experiment design, the multiobjective genetic algorithm and response surface method are combined to optimize the objective functions. The analysis results show that the mass is reduced by 4.35%, the equivalent stress is reduced by 21.05%, the minimum life is increased by 72.28%, and the first-, second-, and third-order natural frequency are also increased to varying degrees. Two different optimization strategies are provided for the design of the drive axle housing.

## 1. Introduction

The drive axle housing of the truck, located at the end of the drive train, is a critical chassis bearing and power transmission component of the vehicle [1]. The drive axle housing mainly bears the reaction force from the ground, as well as the vertical force, longitudinal force, transverse force, and braking torque between the road frame or vehicle body [2]. It requires that the strength and stiffness of the drive axle housing meet its specified service requirements [3]. To realize the lightweight design of a drive axle housing, many scholars have carried out lightweight designs based on different methods.

Yucun Zhou took a light truck drive axle housing as the research object, selected the design parameters, limited stress, and displacement constraints with light weight as the optimization objectives, and realized the purpose of light weight [4]. Bingbing Zhou analyzed the dynamic characteristics and fatigue of the drive axle housing and then carried out the lightweight optimization design to verify the optimized drive axle housing [5]. Yang Chen put forward a reliability analysis method based on the Monte Carlo method by studying the optimization scheme of the drive axle housing of off-road vehicles. On this basis, they carried out a lightweight design and reduced the weight of the drive axle housing by 12.48% [6]. Guo Zhongjia optimized the rear axle housing of a light truck via finite element method based on the mechanical calculation to achieve the lightweight purpose. The low-order-mode natural frequency of the drive axle housing is too high [7]. Zhang Jun optimized it to increase the low-order mode frequency by about 10 Hz under the restriction of mass [8]. Xu Wenchao carried out the Six Sigma robust multiobjective lightweight design of the drive axle housing, studied the influence of wall thickness of the drive axle housing on its performance through the methods of entropy weight and TOPSIS, and combined RBF and NSGA-II algorithms to design a multiobjective lightweight drive axle housing, which significantly improved the performance [9]. Based on static analysis, Yu Yunyun carried out DOE experimental design for 12 design parameters, established the restraint and optimization objective functions, and carried out a multiobjective optimization design, which not only reduced the mass but also guaranteed the performance stability of the drive axle housing [10]. To optimize the space structure of the drive axle housing, Xu Kang carried out the topological optimization design under the limited conditions of different working conditions, which made the stress distribution more uniform while reducing the weight and lowering the maximum stress [11].

Several scholars have studied the dynamic characteristics of the drive axle housing to varying degrees. Wang Xuemei analyzed and optimized the dynamic characteristics of the drive axle housing by the separation method, which improved the minimum fatigue life while reducing the mass [12]. Zheng Bin carried out static and dynamic characteristics analysis and multiobjective optimization design under three different conditions, which improved the comprehensive performance of the drive axle housing [13]. To study the resonance characteristics of the drive axle housing, Li Huilin carried out modal and harmonic response analysis and identified the dynamic characteristics of the drive axle housing [14]. Liu Guozheng analyzed and verified the vibration and noise of the drive axle housing by combining a simulation with a bench test, and obtained the contribution of vibration and noise at different locations [15]. To study the NVH performance of the drive axle, Jiao Dongfeng established a set of NVH performance-analysis methods for the drive axle housing and analyzed the vibration and noise of the passenger car at 60–65 km/h, which verified the validity of the methods [16].

Numerous studies have been carried out to explore the fatigue life of the drive axle housing. As the fatigue failure life of drive axle housing cannot be calculated accurately by simulation analysis or bench test, Shao Yimin put forward a new analysis method based on dynamic strain measurement of actual mine pavement conditions and combined it with finite element analysis to calculate the fatigue failure life, which provides the technique for drive axle housing simulation [17]. Zhao Wei studied the dynamic response and fatigue life prediction of vehicle systems under a random road spectrum and analyzed the magnitude of dynamic stress and fatigue life in the resonance frequency region [18]. To study the vertical fatigue of the drive axle housing, Meng Qinghua proposed a seven-degree-of-freedom dynamic model to predict the fatigue failure under dynamic load and put forward an optimization scheme on this basis [19]. Shang Minzheng used ABAQUS software to carry out dynamic and static characteristics analysis and fatigue analysis of automotive drive axle housing under sinusoidal load spectrum, obtained cloud diagrams of the position and life distribution of failure zone, and proposed improvement plans [20]. Shou Xusong took the old drive axle housing as the research object and conducted a comparative study on residual state and predicted life to achieve high fatigue prediction accuracy [21]. To study the fatigue life uncertainty of the drive axle housing design, Xian Zhongyu carried out static strength analysis and fatigue life analysis by using ABAQUS and researched the cumulative damage and limit safety coefficient [22]. Li Jianmin established a dynamic test system for the drive axle housing of the loader and calculated the fatigue life of the drive axle housing. It was found that the fatigue area is different from the position where the maximum static stress occurs [23]. Fan Zhimin obtained the failure zone of the drive axle housing through transient dynamic analysis and identified the failure zone of the drive axle housing by fatigue analysis. The comparison results between the bench test and the fatigue analysis are consistent [24]. Liu Weiwei carried out dynamic and static characteristics analysis and fatigue life analysis under the maximum vertical force condition of the drive axle housing, extracted the fatigue life and safety factor, and further optimized the design [25]. Yang Zhiqing used HyperMesh software to predict the fatigue life of the drive axle, obtained the distribution of the fatigue life of the drive axle, and verified the rationality of the design [26]. Gao Jing utilized MSC. Patran carried out the finite element analysis of the drive axle, and the simulation results were compared with the fatigue test of the platform to reach a consistent conclusion [27]. Guo Dongqing carried out the modal analysis and optimization of the drive axle, which reduced the mass of the drive axle housing by 27.3% [28].

According to the research carried out by the above scholars, the main research objectives of the drive axle housing are mechanical characteristics, dynamic characteristics, fatigue damage life, and lightweight design with different working conditions. However, due to the uncertainty of the forced vibration of the drive axle housing, the vibration of the drive axle housing is unpredictable. At the same time, different manufacturers and users have different design requirements for the drive axle housing design. To improve the characteristics and design scheme of the drive axle housing, this paper establishes the three-dimensional model of the drive axle through SolidWorks software. The static analysis with four different working conditions is conducted in the ANSYS workbench to obtain the dangerous position and distribution. The fatigue life analysis with the maximum vertical force working condition is carried out to obtain the fatigue distribution of the drive axle housing with alternating loads at different locations. The vibration characteristics and resonant response frequency of the drive axle housing are also analyzed through dynamic characteristics. The random vibration characteristics of the drive axle housing under rough road surfaces are analyzed through random vibration. At the same time, two optimal design schemes are carried out based on mechanical analysis, which are topology optimization design and multiobjective optimization design. Based on the topology optimization and multiobjective optimization method, two lightweight design schemes are provided and two lightweight ideas of drive axle housing are proposed, which provide a reference for the drive axle housing when facing different requirements.

The paper is arranged as follows. In Section 2, the finite element model of the drive axle housing is established. The force analysis diagram of the drive axle housing with four different typical working conditions is given. Section 3 is the finite element analysis of typical working conditions, mainly including the maximum vertical force condition, the maximum traction working condition, the maximum lateral force working condition, the maximum braking working condition, and fatigue life analysis. Section 4 is the dynamic characteristic analysis of the drive axle housing. The vibration characteristics of the drive axle housing are studied through modal analysis, harmonic response analysis, and random vibration response analysis. Section 5 is the optimization design of the drive axle housing using topology optimization and multiobjective optimization to meet different requirements. Section 6 is the conclusions.

## 2. Finite Element Model Setup

### 2.1. Drive Axle Housing Finite Element Model

To study and optimize the static and dynamic characteristics of the drive axle housing of a rear axle drive truck, the solid drive axle housing is modeled in equal proportion to the solid model. Actual restoration of the three-dimensional model used for analysis plays a vital role in simulation, and a better model can better reflect the actual situation under real conditions [29]. Three-dimensional modeling software is used to establish the model of the drive axle housing. The drive axle housing has many small features, such as chamfers and fillets. These small features have little effect on the simulation results. Therefore, in the modeling process, these small features are simplified to facilitate the subsequent meshing and simulation calculation.

Because SolidWorks and ANSYS Workbench can establish the association relationships, there is no need to convert the format of the drive axle housing model file. To avoid errors caused by format conversion after model establishment, the three-dimensional model of the drive axle housing built by SolidWorks can be directly imported into the Workbench through the correlation relationship in the SolidWorks software toolbar.

The three-dimensional solid model of the drive axle housing is meshed in ANSYS. The structural element of solid186 three-dimensional hexahedron with 20 nodes is selected for mesh generation, and the overall size of the mesh is controlled at 15 mm. In the final mesh model, there are 109,416 units and 194,504 nodes, as shown in Figure 1.

When a truck is driving on a rough road surface, the leaf spring seats on both sides of the axle housing bear vertical random dynamic loads. While only the impact load of the axle housing in the vertical direction is considered, and the influence of lateral force and tangential force is not considered, this is the maximum vertical force condition. While the truck runs with the maximum traction, the leaf spring seats of the axle housing bear the vertical load, traction load, and the reaction torque caused by the transmission of driving torque. When the truck is in emergency braking, the leaf spring seats on both sides of the axle housing bear not only the vertical load but also a large emergency braking load. In actual road driving, the drive axle housing often encounters the above working conditions. The load distribution of the drive axle housing is shown in Figure 2.

### 2.2. Selection of Material

The performance of the drive axle housing mainly depends on the mechanical properties of the manufacturing material, which determines the stability of the drive axle housing. The material of drive axle housing studied is SCW550, which has high yield strength and a good economy [10]. The material parameters of SCW550 are shown in Table 1.

## 3. Finite Element Analysis of Typical Working Conditions

Mechanical research on the drive axle housing is carried out to determine the stress concentration area and the maximum deformation area. The analysis results are combined with the actual situation to determine the most critical part of the drive axle housing.

### 3.1. Maximum Vertical Force Working Condition

With the working condition of full-load maximum vertical force, the truck mainly bears the pressure coming from the container and acts on the leaf spring of the truck. The leaf spring acts on the drive axle housing through the leaf spring seat, which will cause the drive axle housing to receive a relatively concentrated working load. At the same time, the vehicle driving on the uneven road surface will bear significant impact loads [30]. At this time, the force in the vertical direction is the largest. For trucks, it is generally 2.5 times the full-load axle load, and the drive axle housing produces bending deformation. The force diagram of the drive axle housing at this working condition is shown in Figure 2a.

The expression of the vertical bending moment between spring seats is shown in Formula (1).
(1)$$M=\left(\frac{G}{2}-g_{w}\right)\frac{B-S}{2}$$

The static bending stress expression is shown in Formula (2).
(2)$$\sigma_{wj}=10^{3}\frac{M}{W_{v}}$$

The expression of bending stress under dynamic load is shown in Formula (3).
(3)$$\sigma_{wd}=k_{d}\sigma_{wj}$$
where *M* is the bending moment between two spring seats, *G* is the rock ground load at rest under full load, *g_w_* is the own wheel gravity, *B* is the spacing between two wheels, *S* is the spacing between two leaf spring seats, *σ_wj_* is the static bending stress, *W_v_* is the vertical bending section factor of the dangerous section, *σ_wd_* is the bending stress under dynamic load, and *k_d_* is the dynamic load factor (here, the *k_d_* is 2.5).

At this working condition, the position where the leaf spring seat is installed on the upper end face of the drive axle housing is subjected to the maximum impact force of 12,500 N in the vertically downward direction. To simulate the actual working condition, the fixed support is used to restrain the half-shaft sleeve at both ends. On this basis, the total deformation and equivalent stress of the drive axle housing with the maximum vertical force are obtained by static analysis. Figure 3a is the total deformation, and Figure 3b is the equivalent stress.

As shown in Figure 3a, the maximum deformation of the drive axle housing is 1.0549 mm, which occurs in the arc transition area between the outer ring of the main drive housing and the half-shaft housing. It decreases in the direction of the half-shaft sleeve. From Figure 3b, it can be seen that the maximum equivalent stress is 205.37 MPa at the step of the half-shaft sleeve, which is also the area where the drive axle housing is prone to damage in practice. Secondly, there are different degrees of stress concentration in the arc transition area and the edge of the main reducer cavity. The stress in the remaining parts is relatively minor.

### 3.2. Maximum Traction Working Condition

When the truck accelerates or climbs a slope, the engine outputs the maximum torque. At this time, the driving wheel will produce the driving force acting on the ground, and the ground will have the same reaction force on the drive axle housing. The drive axle housing is also subjected to the vertical impact force caused by the uneven road surface. These two forces act on the drive axle housing simultaneously, which is the maximum traction working condition. The force diagram of the drive axle housing with the maximum traction working condition is shown in Figure 2b.

The maximum tangential reaction force expression is shown in Formula (4).
(4)$$P_{max}=\frac{T_{max}i_{0}i_{g}\eta_{t}}{r_{0}}$$

The expression of the vertical bending moment between spring seats is shown in Formula (5).
(5)$$M=\left(\frac{G}{2}m_{a}-g_{w}\right)\frac{B-S}{2}$$
where *T_max_* is the maximum engine output torque, *i*_0_ is the transmission ratio of the final drive, *i_g_* is the minimum transmission ratio of the transmission line, *η_t_* is the transmission efficiency, *r*_0_ is the rolling radius of the wheel, and *m_a_* is the mass transfer coefficient of acceleration.

At this working condition, the calculated vertical force is 58,800 N, and the tangential force is 105,586 N, which imposes the same boundary constraints on the drive axle housing. On this basis, the total deformation and equivalent stress of the drive axle housing with the maximum traction condition are obtained by static analysis. Figure 4a is the total deformation, and Figure 4b is the equivalent stress.

From Figure 4a, the maximum deformation of the drive axle housing is 0.50837 mm, which has the same position and distribution rule as the previous working condition and gradually decreases towards the half-shaft sleeve. From Figure 4b, it can be seen that there is a stress concentration at the step position of the half-shaft sleeve with a maximum equivalent stress of 116.53 MPa. The stress concentration area is similar to that under maximum vertical force. The driving axle under acceleration has a small load, so the overall stress is minor, following the actual working conditions.

### 3.3. Maximum Lateral Force Working Condition

When the truck makes a sharp turn, it is subjected to not only a vertical load but also a considerable centrifugal force. With the combined action of the two forces, the truck tends to slip sideways. At the same time, a reaction force will be generated, and the force vectors will work together with the drive axle housing. The forces are appropriately simplified to a vertical and a horizontal one. The force diagram of the drive axle housing with the maximum lateral force working condition is shown in Figure 2c.

The expression of the side slip condition is shown in Formula (6).
(6)$$P_{2}\geq Y_{L}+Y_{R}=G\varphi$$

The expression of the reaction force of the two-wheel supports during side slip is shown in Formula (7).
(7)$$\begin{array}{l} Z_{L}=G\left(\frac{1}{2}-\frac{h_{g}\varphi }{B}\right)\\ Z_{R}=G\left(\frac{1}{2}+\frac{h_{g}\varphi }{B}\right) \end{array}$$
where *P*_2_ is the lateral force acting on the drive axle, *Z_L_* and *Z_R_* are the support reaction on wheels, and *Y_L_* and *Y_R_* are the lateral reaction force acting on the wheels.

It is calculated that the vertical force is 98,000 N, the horizontal force is 98,000 N, and the direction is outward parallel to the half-axis. On this basis, the total deformation and equivalent stress of the drive axle housing with the maximum lateral force working condition are obtained by static analysis. Figure 5a is the total deformation, and Figure 5b is the equivalent stress.

Figure 5a shows that the maximum deformation of the drive axle housing is 0.50177 mm, which occurs at the arc transition of the main reducer chamber of the inner steering race. The distribution trend of the maximum deformation is the same as under the above conditions and decreases towards both ends of the half-shaft sleeve. Figure 5b shows that the maximum equivalent stress of the drive axle housing is 125.7 MPa, located in the step area of the inner-ring half-shaft sleeve. There is a severe stress concentration in this area, located on the side of the inner ring.

### 3.4. Maximum Braking Working Condition

When the truck encounters an emergency, the maximum braking working condition occurs. With this condition, the drive axle housing is not only subjected to vertical load but also the maximum braking force. The force diagram of the driving axle housing with the maximum braking working condition is shown in Figure 2d.

The expression for the vertical bending moment between spring bases is shown in Formula (8).
(8)$$M_{v}=\left(\frac{G}{2}m_{b}-g_{w}\right)\frac{B-S}{2}$$

The expression for the longitudinal bending moment between spring bases is shown in Formula (9).
(9)$$\begin{array}{l} M_{h}=\frac{Gm_{b}\varphi }{2}\times \frac{B-S}{2}\\ m_{b}=1-\frac{h_{g}\varphi }{L_{1}} \end{array}$$

The braking torque expression is shown in Formula (10).
(10)$$T=\frac{G}{2}m_{b}\varphi r_{0}$$
where *m**_b_* is the mass transfer factor when braking, *h_g_* is the truck centroid height, *L*_1_ is the centroid to front axis center distance, and *φ* is the attachment factor (generally 0.75~0.8). The calculated vertical force size is 44,100 N, and the maximum tangential braking force is 3528 N. The above boundary constraints are applied to the half-shaft sleeve.

On this basis, static analysis of the drive axle housing with the maximum braking working condition is carried out to obtain the total deformation and the equivalent stress. Figure 6a is the total deformation, and Figure 6b is the equivalent stress.

As shown in Figure 6a, the maximum deformation of the drive axle housing is 0.38023 mm, which occurs in the same location as the first and second working conditions, and the distribution trend decreases toward both ends of the semi-axle sleeve. As shown in Figure 6b, the maximum equivalent stress-distribution cloud map is similar to the first two working conditions, with a maximum value of 78.974 MPa and a similar distribution of stress-concentration locations.

From the above four working conditions, the maximum deformation occurs at the arc transition of the main decelerator chamber, which is the weak part of the truck drive axle housing. The maximum deformation can be decreased by changing the fillet size of the arc transition. The maximum equivalent stress occurs at the step of the half-shaft casing at all four working conditions, which is the same as the actual damage location. Therefore, the half-shaft sleeve should be strengthened or manufactured with other materials with better mechanical properties to avoid fatigue failure. By comparing the four working conditions given in Table 2, the most dangerous working condition of the truck under the four working conditions is the maximum vertical force working condition, which also belongs to the working condition under which the truck with heavy load bears the most. Therefore, the truck can guarantee good functional stability in this operating condition. Therefore, the subsequent analysis and optimization design are based on the drive axle housing withstanding the maximum vertical force.

### 3.5. Fatigue Life Analysis of Maximum Vertical Force

During the working process, the drive axle housing is continuously subjected to the action of alternating loads, so the primary mode of failure is fatigue failure [31,32]. Fatigue failure is the failure of a part under alternating loads, even under the yield limit of its material. Due to the operating characteristics of the drive axle housing, this working condition is unavoidable. Therefore, the fatigue analysis of the drive axle housing is fundamental. The drive axle housing is an important part, and its design life is 105~107 times, which belongs to high cycle fatigue [33]. Fatigue analysis of the drive axle housing with maximum vertical force is carried out using the fatigue tool in ANSYS Workbench. It is necessary to add the S-N curve of the material and set the fatigue strength factor to 0.8. The drive axle housing bore load history is a non-zero mean random load. The Goodman mean stress theory is used to correct the non-equal amplitude stress and set the type of analysis to stress life [34,35]. The number of cycles is set to 10^9^ in the safety factor analysis module. The S-N curve is calculated based on the material constant of SCW550, as shown in Figure 7.
(11)$$LgN_{i}=a+bLg\sigma_{i}$$
where *a*, *b* is the material constant, and *a* = 22.1144, *b* = −7.1378, *N* represents the number of cycles, and *σ* represents stress.

Finally, the life distribution cloud chart, the safety factor cloud chart, and the damage cloud chart are obtained as shown in Figure 8.

As shown in Figure 8a, the lifetime of the drive axle housing is almost 10^6^ times, which meets the design requirements. There are different degrees of critical positions in the stress-concentration areas of the drive axle housing, which are very prone to fatigue failure. It can be seen from Figure 8b that at the condition of maximum vertical force, the overall safety factor of the drive axle housing is between 1 to 5, the minimum value is 0.33578, and the minimum value appears at the step of the half-shaft sleeve and the arc transition position of the chamber of the main reducer. The minimum value occurs at the stage of the semi-axle casing and the arc transition position of the main decelerator chamber. These are the most dangerous locations, and the overall safety factor is low, so the condition is more dangerous. In manufacturing the drive axle housing, the transition around the main decelerator cavity needs to be strengthened. For the half-shaft sleeve, it needs to be coated or manufactured with metal materials with better mechanical properties. As shown in Figure 8c, the location where fatigue damage is most likely to occur is the same as the location of the truck drive axle housing in the actual working condition, which is located at the step of the axle shaft sleeve.

## 4. Dynamic Characteristic Analysis of the Drive Axle Housing

### 4.1. Modal Analysis of the Drive Axle Housing

The modal analysis of the drive axle housing predicts the dynamic characteristics of the drive axle housing [36,37]. Modal analysis is the best way to obtain the natural frequencies and vibration mode of the system [38]. The first six-order natural frequencies of the modal analysis of the drive axle housing are extracted and compared with the pavement vibration frequencies to determine whether the drive axle housing will produce resonance. The structural dynamic equation is shown in Formula (12).
(12)$$\left[M\right]\left\{\ddot{x}\right\}+\left[C\right]\left\{\dot{x}\right\}+\left[K\right]\left\{x\right\}=\left\{F\left(t\right)\right\}$$
where [*M*] is the structural mass matrix, [*C*] is the system damping matrix, [*K*] is the stiffness matrix or coefficient matrix, {*x*} is the displacement vector, $\left\{\dot{x}\right\}$ is the velocity vector, $\left\{\ddot{x}\right\}$ is the acceleration vector, and {*F*(*t*)} is the excitation force vector.

In the undamped mode analysis, the [*C*] in Formula (12) is reduced to zero, and the external force on the system is zero when the drive axle housing is free to vibrate. Therefore, the dynamic Formula (13) is obtained by setting {*F*(*t*)} to zero in Formula (12).
(13)$$\left[M\right]\left\{\ddot{x}\right\}+\left[K\right]\left\{x\right\}=0$$

For linear systems, the free-vibration displacement solution is shown in Formula (14).
(14)$$\left\{x\right\}=\left\{\Phi_{i}\right\}\cos\omega_{i}t$$
where *ω_i_* is the natural frequency of the i order and *φ_i_* is the mode shape of the *i* order.

The vibration characteristic equation of the structure is shown in Formula (15).
(15)$$(\left[K\right]-\omega_{i}{}^{2}\left[M\right])\left\{\Phi_{i}\right\}=0$$

The truck will inevitably produce vibration when driving on the road. Clearance between transmission systems and long chains will cause vibration, which will cause the drive axle housing to respond. When the road vibration frequency is equal to or multiple of the natural frequency of the drive axle housing, the drive axle housing will resonate. Resonance will not only lead to structural deformation or damage, but also make considerable mechanical noise. It will cause the drive axle housing not to function properly. Therefore, the modal analysis of the drive axle housing is important.

In the modal analysis of the drive axle housing, only the two ends of its half-shaft sleeve are constrained by fixed forces. Because the higher-order modes are not meaningful for practical research, only the first-order to sixth-order modes of the drive axle housing are solved. Finally, the first sixth-order natural frequencies and mode shapes are obtained, as shown in Figure 9.

Table 3 gives the natural frequency and the description of vibration mode of the drive axle housing.

It can be seen from Figure 9 that the frequency of the drive axle housing is mainly concentrated in the range of 70~570 Hz. This range includes the vibration frequency that may occur when driving on the road. It can be seen that the drive axle housing may resonate when driving on the road. The first natural frequency of the drive axle housing is 77.675 Hz, and the maximum deformation occurs in the cavity shell of the drive axle main reducer. The natural frequencies of other orders increases, and the maximum deformation occurs at this position, which is most affected by vibration. This should be considered in the subsequent optimization design process.

### 4.2. Harmonic Response Analysis

For the study of the dynamic characteristics of the drive axle housing, the modal analysis has certain limitations. To further understand the dynamic characteristics of drive axle housing, harmonic response analysis is carried out. Harmonic response analysis is the relationship curve between the dynamic response value and load frequency obtained by determining the excitation load and simple harmonic acting on the linear system. By analyzing the peak value and corresponding frequency in the frequency response curve, it can predict whether the linear structure can overcome the harm caused by forced vibration. Thus, the dynamic characteristics of the drive axle housing can be predicted. The motion equation of the system with simple harmonic load is shown in Formula (16).
(16)$$\left[M\right]\left\{\ddot{x}\right\}+\left[C\right]\left\{\dot{x}\right\}+\left[K\right]\left\{x\right\}=\left\{F\right\}\sin(\theta t)$$
where {*F*} is the amplitude vector of the sinusoidal load and *θ* is the excitation frequency.

The displacement response of its node is Formula (17).
(17)$$\left\{x\right\}=\left\{A\right\}\sin(\theta t+\Phi )$$
where {*A*} is the displacement amplitude vector and Φ is the incentive load displacement response lag phase angle.

Based on the modal analysis, the superposition method is used to solve the harmonic response. The response frequency range is set from 0 to 1000 Hz. To obtain a continuous and stable curve, the solution interval is 400. The surface force is selected as the excitation load to be applied to the leaf spring seat. After the solution, the stepped cylindrical surface of the half-shaft sleeve is chosen as the characteristic surface, and the displacement frequency response curve is derived. Figure 10 shows the displacement frequency response curve of a stepped cylindrical surface of a half-shaft sleeve.

It can be seen from Figure 10 that the frequency distribution of the drive axle housing is susceptible to resonance. It can be seen from Figure 10a that the peak response of the drive axle housing occurs at 120 Hz in the *X*-axis direction with a maximum displacement of 0.013694 mm. Figure 10b shows that the peak response in the *Y*-axis direction is the same as that in the *X*-axis direction, with a frequency of 120 Hz and a maximum displacement of 4.3717 mm. There are several peak responses in the *Z*-axis direction, with the distribution frequencies of 77.5 Hz, 120 Hz, and 337.5 Hz, respectively, with a maximum displacement of 0.0044954 mm. By analyzing the displacement response of the drive axle housing under harmonic excitation load, the peak value is mainly in the range of 120 Hz, and the vibration response in the *Y* direction is the most obvious. The natural frequency of the drive axle housing should be avoided in the design process.

### 4.3. Random Vibration Response Analysis

Road bumps are ubiquitous during driving and can be influenced by other external factors, so the working condition of the truck cannot be estimated and calculated. Therefore, random vibration analysis is critical. The study of random vibration is the response result with statistical meaning. The amplitude and phase of random vibration at any time are unknown, and its vibration mode cannot be formulated by the function, but it has a particular statistical law. Since random vibration loads cannot be represented by time-domain signals, they are represented by frequency-domain signals, called power spectral density [39,40]. Modal analysis should be carried out before random vibration analysis of the drive axle housing. Modal analysis results and the PSD load spectrum should be taken as preprocessing steps for random vibration analysis. The first ten modes are solved, and the natural frequency-distribution folded line is shown in Figure 11. The results of the modal analysis are then transferred to the random vibration module, and PSD G Acceleration dynamic excitation load spectra are added to both ends of the half-shaft sleeve. The direction of the *Y*-axis is set, the load size is 9.8 g^2^/Hz, and the load spectrum frequency range is 1~1000 Hz. Then, the random vibration analysis is carried out.

A confidence interval of 1 sigma is set in the resolver. After the solution of random vibration is completed, the deformation and the equivalent stress cloud chart in the *Y*-axis direction of the drive axle housing are derived.

From Figure 12a, the maximum deformation response value of random vibration of the drive axle housing in a given direction is 0.00016602 mm. The maximum response area is in the arc transition area of the main reducer chamber of the drive axle housing. From Figure 12b, the maximum equivalent stress is 0.089255 MPa, and the distribution of stress concentration appears in the weak position analyzed above. The maximum equivalent stress is at the step of the half-shaft sleeve, but the equivalent stress is minor. Random vibration analysis is based on statistical principles. The statistical error of the simulation value can be determined by comparing the theoretical values obtained from Formula (18) within the confidence interval of the first mode and 1 sigma.

The formula for calculating random vibration for first-order modes and 1-sigma confidence intervals is shown in Formula (18).
(18)$$X={\left[\frac{\pi \gamma^{2}\varphi_{m}{}^{2}PSD}{4\varepsilon M_{m}{}^{2}{\left(2\pi \right)}^{4}f_{0}^{3}}\right]}^{0.5}$$
where *X* is the directional displacement response. *γ* is the mode participation coefficient. *φ_m_* is the maximum mode amplitude of the mode. *ε* is the damping coefficient. *M_m_* is the generalized mass. *f*_0_ is the first order frequency. *PSD* is the power spectrum density, and the *PSD* can be calculated by the following formula.
$$PSD=G_{acceleration}\times 9800\times 9800$$

The displacement response in the *Y*-axis direction within the 1-sigma confidence interval is 0.00011605 mm, which is not significantly different from the simulation value of 0.00016602 mm. Considering the grid refinement, model simplification, and multiple effects of decimal accuracy and magnitude, the simulation results are considered reliable.

## 5. Optimization Design of the Drive Axle Housing

### 5.1. Topology Optimization Design

The mass of the drive axle housing accounts for about 10% of the total mass of the truck. To realize the light weight of the truck, the lightweight design of the drive axle housing should be carried out first. In terms of lightweight design, topological optimization design has apparent advantages, such as optimum allocation of material space ratio and more design freedom [41,42]. Topology optimization guarantees the maximum utilization efficiency of structural materials without significantly impacting performance. For the topology optimization design of the drive axle housing, the optimum area should be determined first. In selecting the optimization area, the matching properties between the drive axle housing and other parts should be comprehensively considered. The optimized structure should not affect the matching properties of other parts. Therefore, the middle position is chosen as the topological optimization area, with the flange discs at both ends of the half-shaft sleeve as the boundary. The solution with the maximum vertical force condition is input into the topology optimization module, and the variable density optimization method is used as the design method to optimize the topology of the drive axle housing.

The mathematical model of the variable density method is shown in Formula (19).
(19)$$\rho =d_{i}\rho_{i}$$
where *ρ* is the unit pseudo-density, *d_i_* is the relative density, and *ρ_i_* is the actual density of each unit.

Based on Formula (19), when *d_i_* is equal to 1, the material should be retained. When *d_i_* is equal to 0, the material is removed. Because the expression of relative density is discrete, it must be continuous in practice. Flexibility minimization is the optimization objective of topology optimization of drive axle housing; that is, to maximize the stiffness of the drive axle housing. The relationship between the two is expressed as Formula (20).
(20)$$C=F^{T}U=U^{T}KU$$
where *C* is the flexibility of the structure, *F* is the load matrix, *U* is the structural displacement matrix, and *K* is the overall stiffness matrix.

The mass retention of the topology optimization of the drive axle housing decreases from 80% to 30% with a gradient of 10%, and a total of six groups of optimization design schemes are obtained. The convergence accuracy of the control solution is 0.1%, the penalty factor is 3, and the maximum number of iterations is 500. The solution is carried out with the condition of maximum vertical force.

The mathematical model of topology optimization is shown in Formula (21).
(21)$$\begin{array}{l} \left\{\begin{array}{l} find\;d={\left[d_{1},d_{2},\cdot \cdot \cdot d_{n}\right]}^{T}\\ min\;C=F^{T}U \end{array}\right.\\ Subject\;to\\ \left\{\begin{array}{l} f=\frac{m-m_{1}}{m}\\ 0\leq d\leq 1\\ F=KU \end{array}\right. \end{array}$$
where *f* is the drive axle housing material retention percentage, *m* is the initial design mass of the drive axle housing, and *m*_1_ is the removal mass of the drive axle housing.

In Figure 13, different colors represent different density distributions. The deletion of drive axle housing materials can be seen from the color distribution characteristics. In the figure, red indicates the low-density area, that is, the area with a relative density close to 0, which represents the deletion of the structural material. Gray represents the high-density area, that is, the area with a relative density close to 1, which represents the retention treatment of the structural material. Yellow is the transition area, which can be selected according to the manufacturing process and processing level of the drive axle housing. Based on the above optimization results, in the actual production, the red removal area needs to be appropriately filled in combination with the work needs, the reserved area should ensure its sufficient stiffness and static strength, and the other regions should be thinned. Most material removal positions are concentrated in the drive axle housing and the main reducer cavity housing. The smaller the mass retention percentage is, the more material is removed from the main reducer cavity shell. Most material removal positions are located at the position with less equivalent stress, which is opposite to the distribution of the stress cloud chart. Select the optimization results with 50% quality retention for the verification and analysis of the optimization results.

Based on the material removal strategy given by topology optimization, the model is modified by following methods.

The upper and lower end faces of the main reducer chamber

Step 1: The thickness of the mounting end of the leaf spring seats is reduced by 20 mm, and the fillet is 5 mm.

Step 2: A cylinder with a diameter of 176 mm and a height of 11.25 mm is cut from the upper and lower surfaces of the main reducer chamber, and the fillet is 60 mm.

Step 3: The fillet of the transition area of the main reducer chamber is 60 mm.

The modified model is shown in Figure 14.

The statics analysis of the optimized drive axle housing is carried out again, and the results are shown in Figure 15. 

After topology optimization, the maximum deformation increases slightly compared to before optimization; the value is 1.2283 mm. The maximum equivalent stress increased by 24.17 MPa to 229.54 MPa compared with that before optimization. Compared with the initial model, the stress and deformation are increased. This is because there are certain errors in the manual processing of the model, and the material cannot be deleted completely according to the topology optimization results, resulting in a slight increase in the deformation stress of the model. The distribution is the same as that before optimization.

The fatigue analysis of the optimized drive axle housing with maximum vertical force is carried out again using the fatigue tool in ANSYS, and the results are shown in Figure 16.

After optimization, the minimum life and minimum safety factor decreased to varying degrees. The minimum life decreased from 10,723 to 7551.5, a reduction of 29.5%. The safety factor decreased from 0.33578 to 0.30043, a reduction of 10.5%.

Then, the modal analysis of the optimized drive axle housing is carried out, and the results are shown in Table 4 and Figure 17. 

After optimization, the natural frequency changes little, the vibration types are similar, the maximum deformation increases, the sixth-order frequency drops close to 80 Hz, and the equivalent stress and fatigue life have certain changes. However, they are all within the allowable range. The mass of the drive axle housing decreased significantly, from 504.88 Kg to 416.85 Kg, a decrease of 17.4%. The purpose of light weight is achieved.

### 5.2. Multiobjective Optimization Design

The drive axle housing is a critical component in the automobile chassis. When optimizing the drive axle housing, its static and dynamic characteristics will be affected, and the optimization results of favoring one over the other will further jeopardize the stability and safety of the vehicle. Based on this background, a multiobjective optimization design method based on the response surface method is adopted. Due to the coupling relationship between the structural parameters of the drive axle housing, the size to be optimized is taken as the design variable, the variable range is given, and the optimization objective is taken as the objective function. Within the given variable range, the best position between the given objective functions is searched through the changes in the design variables [43,44]. The multiobjective size-optimization design of the drive axle housing is to seek the optimal solution of the objective function on the premise of ensuring its functional performance. While meeting the lightweight requirement, it can also meet other performance requirements.

Due to the assembly relationship between the half-axle sleeve and the truck, the change in its size may lead to the change of mating parts. At the same time, the mass of the axle shaft sleeve is small in the whole drive axle housing, so it should be eliminated when optimizing the drive axle housing. Therefore, the multiobjective size-optimization design is only carried out for the middle chamber of the half-shaft sleeve, and the optimization design parameters are shown in Figure 18.

When carrying out a multiobjective optimization design of drive axle housing based on the response surface method, DOE experimental design is required first. The range of design parameters is shown in Table 5.

During DOE experimental design, the experimental method is selected as the optimal space-filling design method. This method can minimize the number of sample points, accurately reflect the changes of design points, and significantly avoid the inaccurate experimental results caused by too concentrated sample points [45,46,47]. The sample type is CCD, and 27 samples are obtained. The mass, deformation, equivalent stress, service life, and the first, second, and third natural frequency of the drive axle housing with the maximum vertical force are taken as the optimization objective functions, expressed in *y*_1_~*y*_7_, respectively.

Through the response surface method, the corresponding tests are carried out on the test points, and the test results are mathematically analyzed to establish the response surface model. The relationship between design parameters and the objective function is observed through the response surface [48,49,50]. The mathematical expression of the response surface method is shown in Formula (22).
(22)$$y=\beta_{0}+\sum_{i=1}^{k}\beta_{i}x_{i}+\sum_{i=1}^{k}\beta_{ii}x_{i}{}^{2}+\sum_{i=1}^{k-1}\sum_{j=i+1}^{k}\beta_{ij}x_{i}x_{j}+\zeta$$
where *y* is the dependent variable; *x_i_* and *x_j_* are design variables; *i* = 1, 2, 3, … *k*; *k* is the total number of design variables; *β*_0_, *β_i_*, *β_ii_*, and *β_ij_* are the response surface regression coefficients; *ζ* is the response surface-fitting error.

According to the data obtained from DOE experimental design, multiple quadratic regression equation fitting is carried out to obtain the response value between design parameters and design points. The quadratic response surface mathematical model is established, as shown in Formulae (23)–(29).
(23)$$\begin{array}{ll} y_{1}= & 505.63+10.98x_{1}+6.19x_{2}+4.42x_{3}+69.44x_{4}-4.83x_{5}+3.82x_{1}x_{4}\\ & +2.64x_{2}x_{4}-1.89x_{2}{}^{2}-1.48x_{3}{}^{2}-16.79x_{4}{}^{2}-1.57x_{5}{}^{2} \end{array}$$
(24)$$\begin{array}{ll} y_{2}= & 1.05-8.38\times 10^{-2}x_{1}-7.01\times 10^{-3}x_{2}-5.20\times 10^{-2}x_{3}-1.61\times 10^{-1}x_{4}+7.61\times 10^{-3}x_{5}\\ & +1.09\times 10^{-2}x_{1}x_{4}+6.53\times 10^{-3}x_{1}x_{5}-5.96\times 10^{-3}x_{2}x_{5}+1.12\times 10^{-2}x_{3}x_{4}+1.32\times 10^{-2}x_{2}{}^{2}\\ & +9.71\times 10^{-3}x_{3}{}^{2}+8.56\times 10^{-2}x_{4}{}^{2}+8.51\times 10^{-3}x_{5}{}^{2} \end{array}$$
(25)$$\begin{array}{ll} y_{3}= & 130.09-18.66x_{1}-13.88x_{2}-6.92x_{3}-18.90x_{4}+2.78x_{5}\\ & +47.33x_{1}{}^{2}+45.47x_{2}{}^{2}+56.56x_{3}{}^{2}+54.40x_{4}{}^{2}+49.50x_{5}{}^{2} \end{array}$$
(26)$$y_{4}=9958.40+3019.74x_{1}+1889.27x_{2}+2826.11x_{4}+2290.14x_{1}x_{2}$$
(27)$$y_{5}=75.57+0.54x_{1}+5.46x_{4}+0.04x_{5}-1.72x_{1}x_{5}-1.50x_{4}{}^{2}$$
(28)$$\begin{array}{ll} y_{6}= & 120.45+4.87x_{1}-0.15x_{2}+1.13x_{3}+1.60x_{4}+0.14x_{5}-0.44x_{1}x_{4}\\ & -0.40x_{1}x_{5}+0.29x_{2}x_{5}-0.54x_{2}{}^{2}-0.34x_{3}{}^{2}-1.07x_{4}{}^{2}-0.35x_{5}{}^{2} \end{array}$$
(29)$$\begin{array}{ll} y_{7}= & 168.19+3.76x_{1}+0.29x_{2}+0.94x_{3}+2.61x_{4}+0.06x_{5}-1.17x_{2}{}^{2}\\ & -0.96x_{3}{}^{2}-5.86x_{4}{}^{2}-1.31x_{5}{}^{2} \end{array}$$

The obtained response surface model is shown in Figure 19.

Through the orthogonal experimental design and analysis of DOE data, the relationship curve between the design parameters of the drive axle housing and the objective function is obtained by the response surface method. It can be seen from Figure 19a that *x*_1_ has a weak impact on *y*_1_, and the parameter that mainly affects *y*_1_ response value is *x*_4_, which shows a positive correlation with it. Figure 19b shows that the design parameter that has a significant impact on *y*_2_ is *x*_4_, showing a positive correlation. As shown in Figure 19c, *y*_3_ is affected by *x*_2_ and *x*_3_ with little difference. It can be seen from Figure 19d that *x*_1_ and *x*_2_ have a positive correlation with *y*_4_, and the influence of *x*_1_ is greater than that of *x*_2_. Figure 19e shows that the effect of *x*_1_ and *x*_5_ on *y*_5_ is in the opposite relationship, and the effect alone is a negative correlation. Figure 19f shows that *x*_2_ has little impact on *y*_6_, and *x*_1_ has a positively correlates with y6. Figure 19g shows that *x*_3_ has little impact on *y*_7_, and *x*_4_ increases first and then decreases with *y*_7_.

Multiobjective optimization sets the optimization expectation for the design points under the constraints of design parameters and given boundary conditions. The analysis system designs according to the optimization expectation. It makes the design of the drive axle housing meet the expected requirements of each design point to the greatest extent through mathematical or statistical methods.

The multiobjective optimization mathematical model of the drive axle housing is as follows:(30)$$\begin{array}{l} \left\{\begin{array}{l} Find\;X={\left[x_{1},x_{2},x_{3},x_{4},x_{5}\right]}^{T}\\ min\left[y_{1},y_{2},y_{3}\right]\\ max\left[y_{4},y_{5},y_{6},y_{7}\right] \end{array}\right.\\ Subject\;to\\ \underline{x}\leq x\leq \bar{x} \end{array}$$
where *X* is the five design parameter points shown in Table 5; $\underline{x}$ is the lower limit of the design parameters; $\bar{x}$ is the upper limit of the design parameters; *min* is used to find the minimum value; *max* is used to find the maximum value; *y**_i_* is the objective function; *i* = 1, 2, 3,…, 7.

The optimization boundary conditions expressed in the above model are used for the solution, and the multiobjective genetic algorithm based on the NSGA-II algorithm is used for the solution. Three groups of optimal candidate points are obtained, as shown in Table 6.

The above three groups of results are the optimal design solution, which can be selected according to the actual needs in the manufacturing process. After analysis, the first group is selected as the optimization result and compared with the initial data, as shown in Table 7.

By comparing the data in the table, the mass of the optimized drive axle housing is reduced from 504.88 Kg to 482.92 Kg, a decrease of 4.35%. The deformation increases slightly from 1.06 mm to 1.09 mm, but it is negligible for the actual work of the drive axle housing. The maximum equivalent stress decreased from 205.37 MPa to 162.14 MPa, a decrease of 21.05%. The service life increased significantly from 10,723 to 18,474.10, an increase of 72.28%. The first, second, and third natural frequencies are increased in varying degrees, of which the maximum increase percentage is 4.25%, from 77.68 Hz to 80.98 Hz, and the increase in other orders is slight, which are 1.21% and 0.67%, respectively. According to the data analysis and comparison, the drive axle housing not only realizes the light weight but also ensures the performance of the drive axle housing. Compared with the topology optimization results, although the degree of lightweight is slightly poor, it has no impact on the performance of the drive axle housing. The selection of the two optimization schemes should be made according to the needs of designers.

## 6. Conclusions

(1)The dynamic and static characteristics of the drive axle housing are studied. Through the ANSYS Workbench static simulation analysis of the drive axle housing model, the equivalent stress and deformation cloud charts of the drive axle housing with four working conditions are obtained, and the distribution of stress and deformation with four different working conditions is identified.(2)The fatigue analysis of the maximum vertical force condition is carried out to find the location where fatigue damage can easily occur.(3)The first six natural frequencies of the drive axle housing are obtained through modal analysis. Compared with the road driving vibration frequency, it is found that the drive axle housing produces driving resonance easily.(4)Based on the modal analysis, the harmonic response analysis and random vibration analysis are carried out, the resonance-prone frequency points are extracted, and the displacement response value and global equivalent stress distribution in a given direction are obtained.(5)Through the analysis of the dynamic and static characteristics of the drive axle housing, it is found that the most prone stress-concentration position of the drive axle housing is at the step of the half-shaft sleeve, which is consistent with the actual failure position.(6)Through two different optimization methods, the experimental optimization design of the drive axle housing is carried out. The topology optimization reduces the mass by 17.4%, but the performance of the drive axle housing decreases slightly.(7)Finally, the multiobjective optimization design is carried out. The DOE experiment is carried out through the optimal space-filling design method. The response surface method is used, and the response surface model is established. Finally, the multiobjective genetic algorithm is used to optimize the design parameters.(8)The optimization results meet the requirements of light weight and optimal performance of the drive axle housing. The paper provides two different optimization methods for the design of the drive axle housing.

## Figures and Tables

**Figure 1 materials-15-05268-f001:**
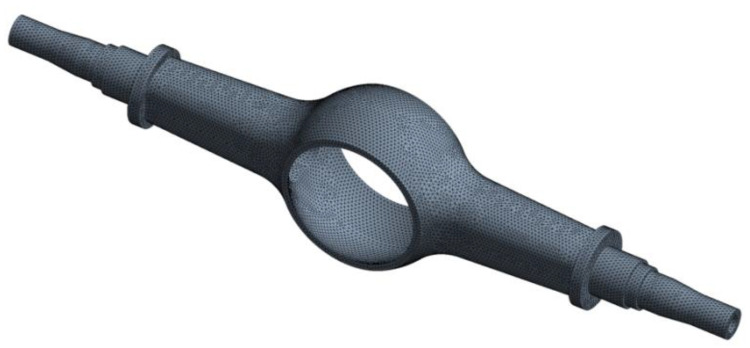
Mesh model of the drive axle housing.

**Figure 2 materials-15-05268-f002:**
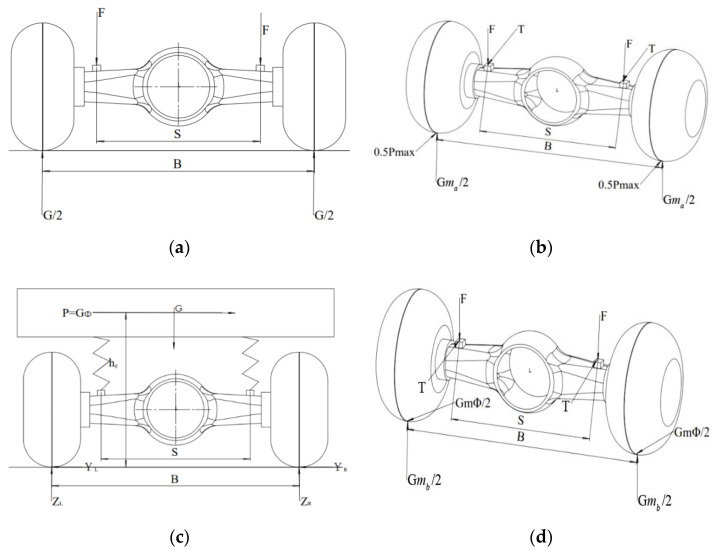
Force analysis diagram of the drive axle housing with different working conditions: (**a**) force diagram of the drive axle housing with maximum vertical force condition; (**b**) force diagram of the drive axle housing with the maximum traction working condition; (**c**) force diagram of the drive axle housing with the maximum lateral force working condition; (**d**) force diagram of the driving axle housing with the maximum braking working condition.

**Figure 3 materials-15-05268-f003:**
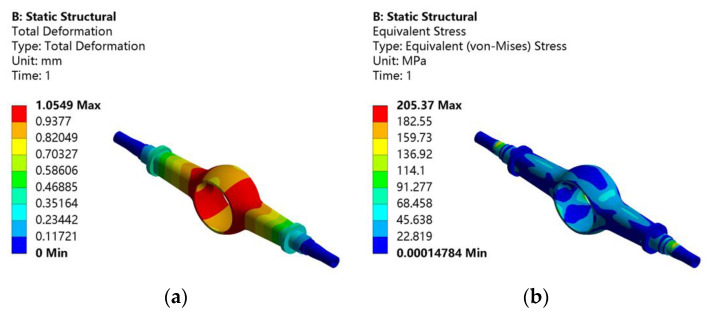
Static analysis results of the drive axle housing with the maximum vertical force condition: (**a**) total deformation; (**b**) equivalent stress.

**Figure 4 materials-15-05268-f004:**
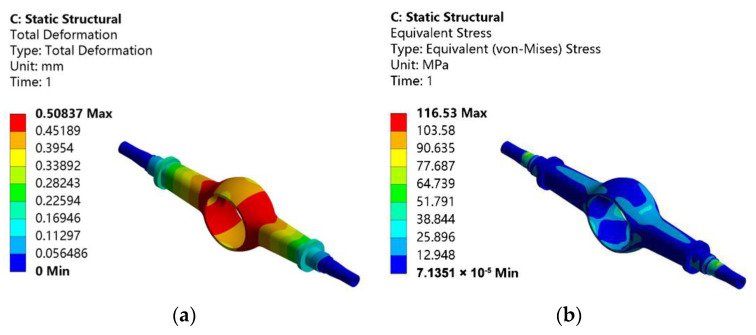
Static analysis results of the drive axle housing with the maximum traction working condition: (**a**) total deformation; (**b**) equivalent stress.

**Figure 5 materials-15-05268-f005:**
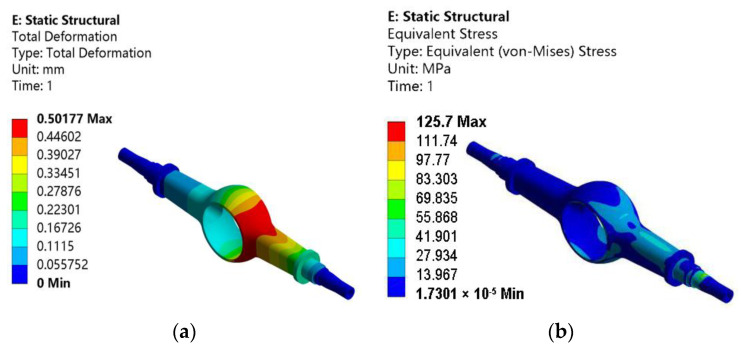
Static analysis results of the drive axle housing with the maximum lateral force condition: (**a**) total deformation; (**b**) equivalent stress.

**Figure 6 materials-15-05268-f006:**
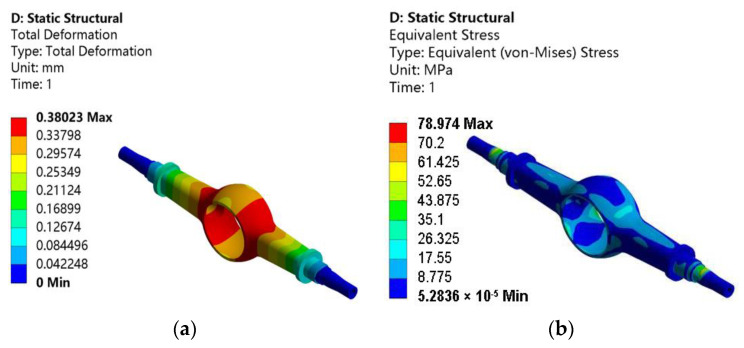
Static analysis results of the drive axle housing with maximum braking working condition: (**a**) total deformation; (**b**) equivalent stress.

**Figure 7 materials-15-05268-f007:**
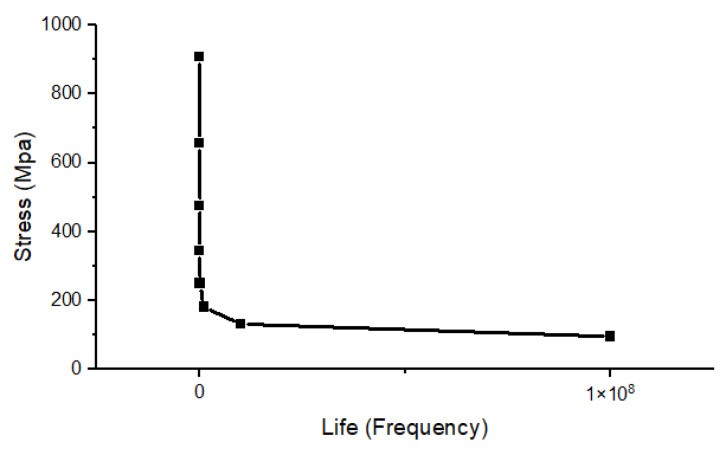
S-N curve of SCW550.

**Figure 8 materials-15-05268-f008:**
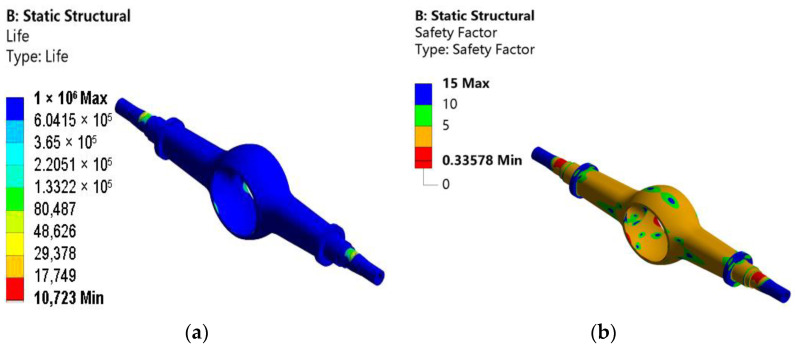
Cloud chart of the fatigue life analysis: (**a**) fatigue life; (**b**) safety factor; (**c**) distribution of damage.

**Figure 9 materials-15-05268-f009:**
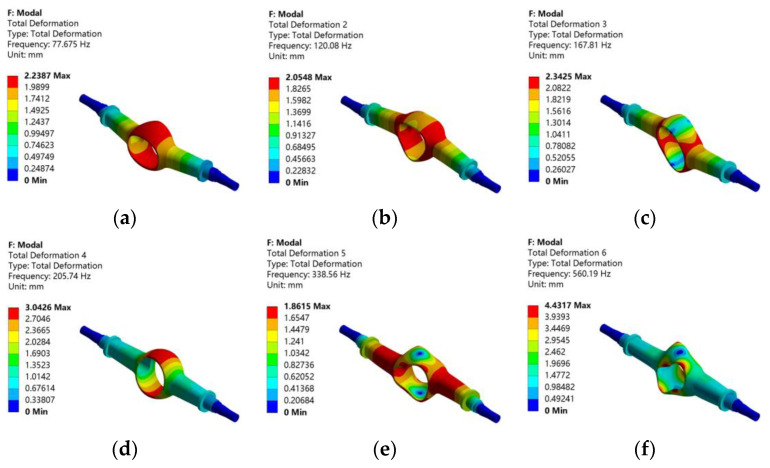
The first-order to sixth-order vibration modes of the drive axle housing: (**a**) first-order mode; (**b**) second-order mode; (**c**) third-order mode; (**d**) fourth-order mode; (**e**) fifth-order mode; (**f**) sixth-order mode.

**Figure 10 materials-15-05268-f010:**
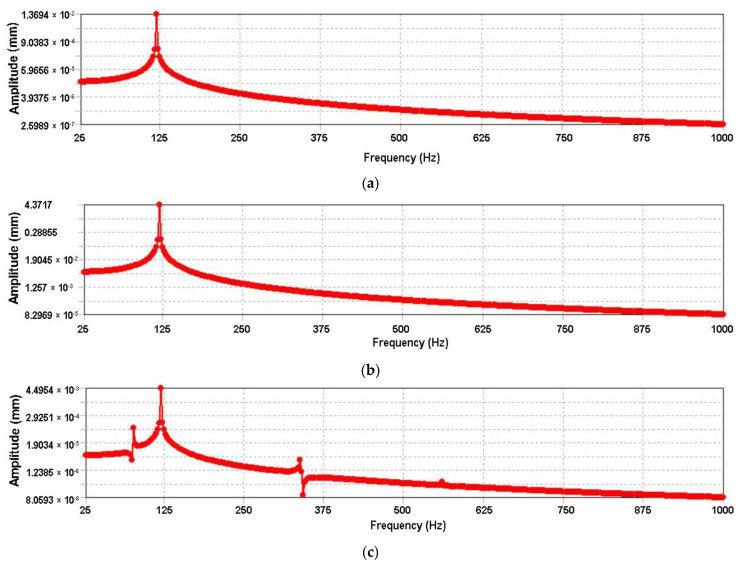
Displacement frequency response curve: (**a**) *X*-axis-direction displacement frequency response curve; (**b**) *Y*-axis-direction displacement frequency response curve; (**c**) *Z*-axis-direction displacement frequency response curve.

**Figure 11 materials-15-05268-f011:**
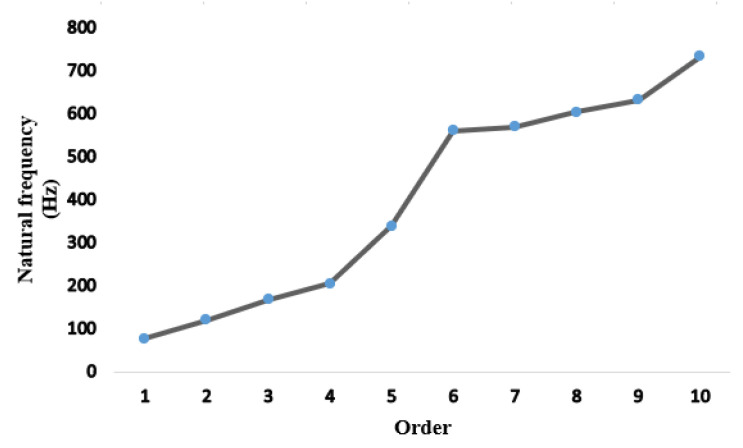
Chart of first- to tenth-order natural frequency distribution.

**Figure 12 materials-15-05268-f012:**
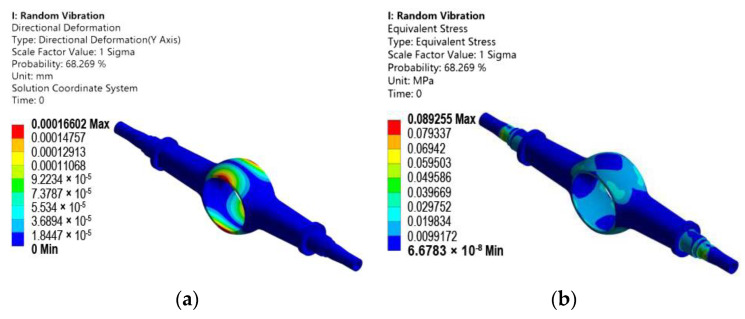
Results of random vibration analysis: (**a**) *Y*-axis-direction deformation cloud chart; (**b**) *Y*-axis-direction equivalent stress cloud chart.

**Figure 13 materials-15-05268-f013:**
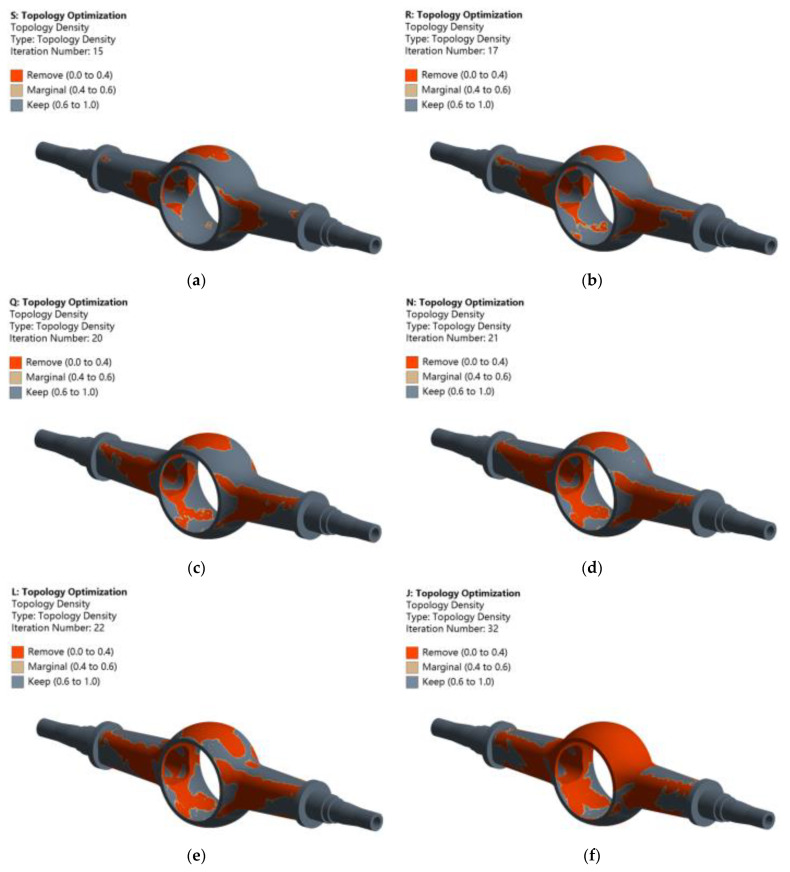
Topology optimization pseudo-density cloud chart: (**a**) retains 80% mass; (**b**) retains 70% mass; (**c**) retains 60% mass; (**d**) retains 50% mass; (**e**) retains 40% mass; (**f**) retains 30% mass.

**Figure 14 materials-15-05268-f014:**
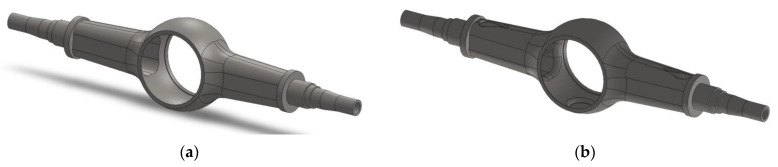
Comparison of optimized models: (**a**) before optimization; (**b**) after optimization.

**Figure 15 materials-15-05268-f015:**
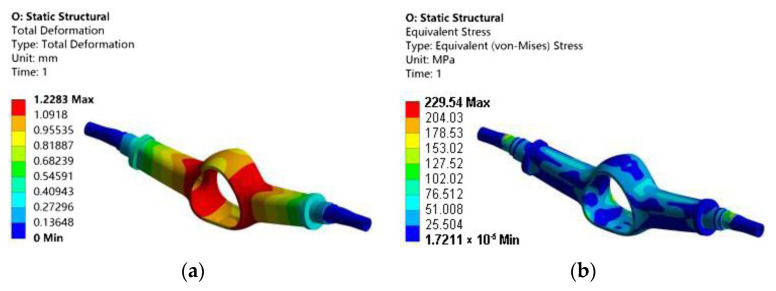
Cloud chart of static analysis after optimization: (**a**) total deformation; (**b**) equivalent stress.

**Figure 16 materials-15-05268-f016:**
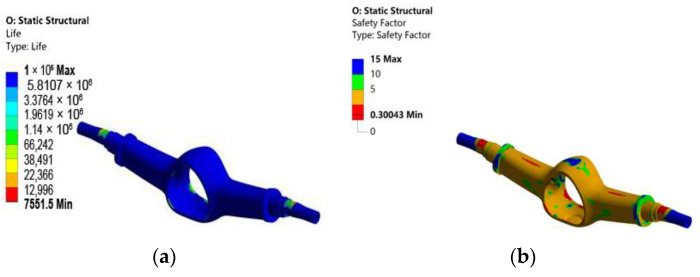
Fatigue life results of the optimized drive axle housing: (**a**) life cloud chart; (**b**) safety factor cloud chart.

**Figure 17 materials-15-05268-f017:**
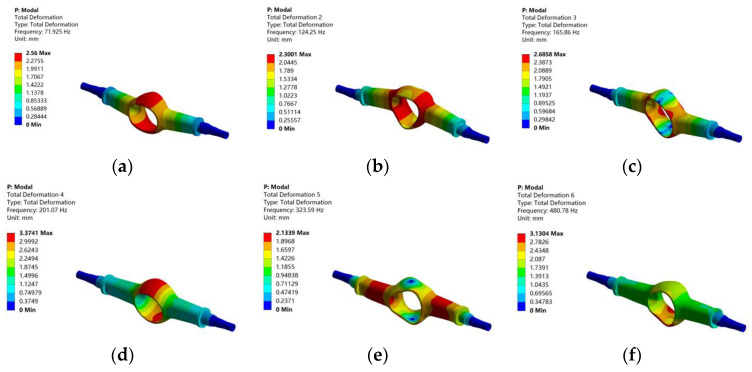
The cloud chart of the first-order to sixth-order vibration modes of the optimized drive axle housing: (**a**) first-order mode; (**b**) second-order mode; (**c**) third-order mode; (**d**) fourth-order mode; (**e**) fifth-order mode; (**f**) sixth-order mode.

**Figure 18 materials-15-05268-f018:**
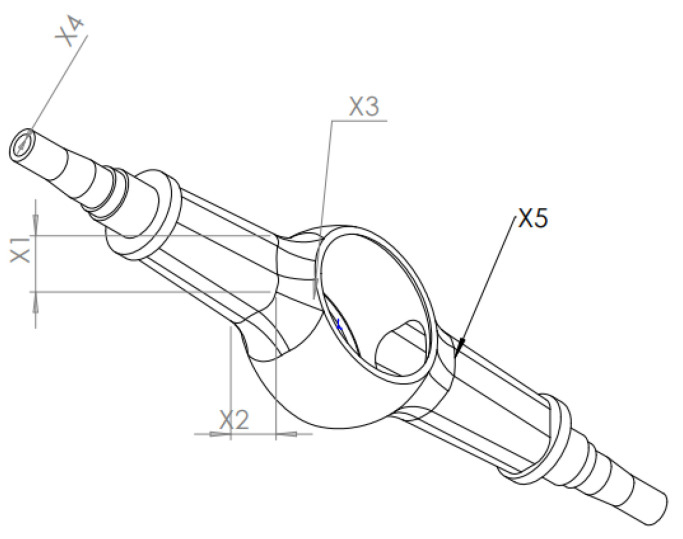
Schematic diagram of design parameters.

**Figure 19 materials-15-05268-f019:**
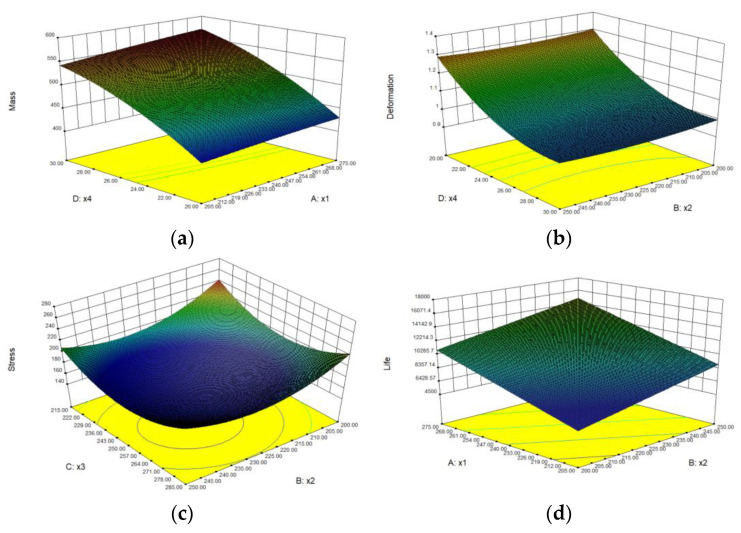
Response surface model diagram: (**a**) response surface model of *x*_1_ and *x*_4_ to *y*_1_; (**b**) response surface model of *x*_2_ and *x*_4_ to *y*_2_; (**c**) response surface model of *x*_2_ and *x*_3_ to *y*_3_; (**d**) response surface model of *x*_1_ and *x*_2_ to *y*_4_; (**e**) response surface model of *x*_1_ and *x*_5_ to *y*_5_; (**f**) response surface model of *x*_1_ and *x*_2_ to *y*_6_; (**g**) response surface model of *x*_3_ and *x*_4_ to *y*_7_.

**Table 1 materials-15-05268-t001:** Material parameters of SCW550.

Material	Yield Strength/MPa	Tensile Strength/MPa	Elastic Modulus/MPa	Poisson Ratio	Density/kg·m^−3^
SCW550	420	570	212,000	0.31	7850

**Table 2 materials-15-05268-t002:** Static analysis results of the drive axle housing with four working conditions.

Working Condition	Maximum Equivalent Stress/MPa	Maximum Deformation/mm
Maximum vertical force working condition	205.37	1.0549
Maximum traction working condition	116.53	0.50837
Maximum lateral force working condition	125.7	0.50177
Maximum braking working condition	78.974	0.38023

**Table 3 materials-15-05268-t003:** Description of the first-order to sixth-order natural frequencies and vibration types of the drive axle housing.

Order	Natural Frequency/Hz	Description of Vibration Mode
1	77.675	The drive axle housing swings in the *Z*-axis direction.
2	120.08	The drive axle housing swings in the *Y*-axis direction.
3	167.81	The drive axle housing is twisted in the XOY plane.
4	205.74	The drive axle housing is twisted in the *X*-axis direction.
5	338.56	The drive axle housing is twisted in the *Y*-axis direction.
6	560.19	The drive axle housing twists and swings in space.

**Table 4 materials-15-05268-t004:** The first-order to sixth-order natural frequency and description of vibration mode of the optimized drive axle housing.

Order	Natural Frequency/Hz	Description of Vibration Mode
1	71.925	The drive axle housing swings in the *Z*-axis direction.
2	124.25	The drive axle housing swings in the *Y*-axis direction.
3	165.86	The drive axle housing is twisted in the XOY plane.
4	201.07	The drive axle housing is twisted in the *X*-axis direction.
5	323.59	The drive axle housing is twisted in the *Y*-axis direction.
6	480.78	The drive axle housing twists and swings in space.

**Table 5 materials-15-05268-t005:** Design parameter range and initial value.

Design Parameter	Design Parameter Name	Initial Value of Design Parameters/mm	Design Parameter Range/mm
*x* _1_	Half-shaft cavity height	238	205~275
*x* _2_	Half-shaft cavity width	225	200~250
*x* _3_	Transition fillet between axle shaft Cavity and main reducer cavity	250	215~285
*x* _4_	The thickness of the drive axle housing	25	20~30
*x* _5_	Drive axle half-shaft cavity fillet	75	60~90

**Table 6 materials-15-05268-t006:** Three sets of optimal solutions.

	Group 1	Group 2	Group 3
*x*_1_/mm	247.64	252.37	245.62
*x*_2_/mm	233.10	233.17	235.23
*x*_3_/mm	252.70	250.78	249.91
*x*_4_/mm	23.10	23.13	23.15
*x*_5_/mm	73.99	73.70	73.77
*y*_1_/kg	482.92	484.49	483.25
*y*_2_/mm	1.09	1.08	1.10
*y*_3_/MPa	162.14	161.61	162.84
*y* _4_	18,474.10	18,768.41	18,667.21
*y*_5_/Hz	80.98	80.82	80.88
*y*_6_/Hz	121.53	122.13	121.10
*y*_7_/Hz	168.93	169.39	168.62

**Table 7 materials-15-05268-t007:** Comparison of results before and after optimization.

Design Parameters	Before Optimization	After Optimization	After Rounding
*x*_1_/mm	238	247.64	248
*x*_2_/mm	225	233.10	233
*x*_3_/mm	250	252.70	253
*x*_4_/mm	25	23.10	23
*x*_5_/mm	75	73.99	74
*y*_1_/kg	504.88	482.92	482.92
*y*_2_/mm	1.06	1.09	1.09
*y*_3_/MPa	205.37	162.14	162.14
*y* _4_	10,723	18,474.10	18,474.10
*y*_5_/Hz	77.68	80.98	80.98
*y*_6_/Hz	120.08	121.53	121.53
*y*_7_/Hz	167.81	168.93	168.93

## Data Availability

The data of numerical simulation used to support the findings of this study are included within the article.

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
