# Peer review of "Topology Optimization and Multiobjective Optimization for Drive Axle Housing of a Rear Axle Drive Truck"

_materials, 2022, doi:10.3390/ma15155268_

Round 1

Reviewer 1 Report

Thank you for submitting your paper. The work done here draws attention to a significant subject on design of housing of drive axel. I have found the paper to be interesting. However, several issues need to be addressed properly before the paper is being considered for publication. My comments including major and minor concerns are given below:

Please consider reviewing the abstract and highlight the novelty, major findings, and conclusions. I suggest reorganizing the abstract, highlighting the novelties introduced. The abstract should contain answers to the following questions:

What problem was studied and why is it important?

What methods were used?

What conclusions can be drawn from the results? (Please provide specific results and not generic ones).

The abstract must be improved. It should be expanded. Please use numbers or % terms to clearly shows us the results in your experimental work.

Please consider reporting on studies related to your work from mdpi journals.

The authors should add a list of nomenclature for all the Greek letters and symbols used in the study.

2 Drive axle housing finite element model change the title to something better, for example “Finite element model setup or something else similar.

Title of the manuscript should be improved, for example “Topology Optimization and Multi Objective Optimization of drive axle housing of a rear axle drive truck”

 Combine figures 1 and 2 and use (a) and (b) instead.

Figure 1 is somewhat useless, it does not add any value or useful info to the paper, the authors need to add some additional info to the figure such as dimensions (2D view)

Where did the authors get the data for Table 1?

The FE model lacks so many details such as element type, convergence study, number of elements, boundary conditions and so on…authors need to provide complete details regarding this issue.

3 Static analysis with different working conditions this is not a good title for a section, author either delete and combine with previous section or improve it.

Lines 171-176 generic info which is more suited into introduction and not here. Authors should consider moving it or deleting it.

The authors need to add a full list of nomenclature for all the Greek letters, math symbols and abbreviations at the end or beginning of the manuscript.

It is not clear where the results and discussion chapter starts, the authors need to follow journal guidelines and recommendations for organising the manuscript layout.

Figure 4 is missing units, authors should add them and check for other figures as well

Figures 3, 5, 7 and 9 should be combined into one larger figure and probably move it to the modelling section 2 to describe the case scenarios.

Since there is a symmetry in the model, why the authors did not take advantage of that and model ½ of the axel instead of all it.

Line 371 instead of using words such as “dangerous” the authors should use words such as “critical”

The results are merely described and is limited to comparing the numerical observation and describing results using FE methods. The authors are encouraged to include a more detailed results and discussion section and critically discuss the observations from this investigation with existing literature.

Conclusions can be written in bullet points for each of the subsections in the results and discussion section.

what is the percentage of self citations in the paper?

I don’t see where is the topology optimization? Is it based on mass reduction? If yes then were did the authors remove materials from, where is the updated topology optimized model?

Author Response

Dear reviewer,

On behalf of my co-authors, we thank you very much for giving us an opportunity to revise our manuscript. We appreciate editor and reviewers very much for their positive and constructive comments and suggestions on our manuscript.

We have studied reviewer’s comments carefully and have made revision in the paper. We have tried our best to revise our manuscript according to the comments. Attached please find the revised version, which we would like to submit for your kind consideration.

We would like to express our great appreciation to editor and reviewers for comments on our paper. Looking forward to hearing from you.

Please check the attachement for all revisions of the paper.

Kind regards,

Reviewer 2 Report

The study proposed a framework for optimization of topology optimization design of drive axle housing. The results show the feasibility of topology optimization for the design of the drive axle housing which are very important and useful to reduce the mass of drive axle housing. The paper includes new contributions with good merits for publication. However, a number of issues/errors in the manuscript are expected to be solved.

- The novelty of the paper should be properly addressed in abstract, the introduction and the conclusion.

- I think that the first sentence of the introduction ( The introduction should …. Important) should be deleted.

- The first paragraph in the introduction is confusing because it contains duplicated sentences; lines from 33 to 39 are the same as lines from 39 to 46.

- The review of optimization methods can be expanded, some RBDO works are recommended to be added, such as:

Reliability based geometrically nonlinear bi-directional evolutionary structural optimization of elasto-plastic material. Scientific Reports, 12(1), 1-22.  Doi: 10.1038/s41598-022-09612-z

Elasto-Plastic limit analysis of reliability based geometrically nonlinear bi-directional evolutionary topology optimization. Structures 34(December 2021):1720-1733. 10.1016/j.istruc.2021.08.105

- It will be more appropriate if you add a paragraph at the end of the introduction section illustrating the layout of the paper.

- Sections 2 and 3 should reorganized to have subsections.

- Please follow the instructions of the journal to prepare the paper (e.g., the font style and size, citation style … etc.).

Author Response

Modification Report

Dear reviewer,

On behalf of my co-authors, we thank you very much for giving us an opportunity to revise our manuscript. We appreciate editor and reviewers very much for their positive and constructive comments and suggestions on our manuscript.

We have studied reviewer’s comments carefully and have made revision in the paper. We have tried our best to revise our manuscript according to the comments. Attached please find the revised version, which we would like to submit for your kind consideration.

We would like to express our great appreciation to editor and reviewers for comments on our paper. Looking forward to hearing from you.

All the modifications of the paper as follows,

Reviewer:

(1)The novelty of the paper should be properly addressed in abstract, the introduction and the conclusion.

Answer: Yes, we have carefully reviewed and revised the abstract, introduction and conclusion, and embodied the novelty in them.

(2) I think that the first sentence of the introduction ( The introduction should …. Important) should be deleted.

Answer: Yes, the first sentence of the introduction has been deleted.

   (3)The first paragraph in the introduction is confusing because it contains duplicated sentences; lines from 33 to 39 are the same as lines from 39 to 46.

  Answer: Due to the error in processing the article when submitting, the introduction is repeated, and now the repeated part has been deleted.

(4) The review of optimization methods can be expanded, some RBDO works are recommended to be added, such as:‎

Reliability based geometrically nonlinear bi-directional evolutionary structural optimization of elasto-plastic material. Scientific Reports, 12(1), 1-22.  Doi: 10.1038/s41598-022-09612-z

Elasto-Plastic limit analysis of reliability based geometrically nonlinear bi-directional evolutionary topology optimization. Structures 34(December 2021):1720-1733. 10.1016/j.istruc.2021.08.105

 Answer: Yes, we have downloaded and added these two papers. At the same time, these papers are very valuable, which we have carefully read and cited as the reference basis of our article.

(5) It will be more appropriate if you add a paragraph at the end of the introduction section illustrating the layout of the paper.

  Answer: Yes, we have added the layout description of the paper at the end of the introduction according to the layout of the paper as following:

The paper is arranged as follows. In Section 2, the finite element model of the drive axle housing is established. The force analysis diagram of the drive axle housing with four different typical working conditions is given. Section 3 is the finite element analysis of typical working conditions, mainly including the maximum vertical force condition, the maximum traction working condition, the maximum lateral force working condition, the maximum braking working condition and fatigue life analysis. Section 4 is dynamic characteristic analysis of the drive axle housing. The vibration characteristics of the drive axle housing are studied through modal analysis, harmonic response analysis and random vibration response analysis. Section 5 is optimization design of the drive axle housing using topology optimization and multi-objective optimization to meet different requirements. Section 6 is the conclusions.

(6)Sections 2 and 3 should reorganized to have subsections.

Answer: The second and third sections have been modified to move the force diagram of the driving axle housing from the third section to the second section. The title of the third section has also been modified.

(7)Please follow the instructions of the journal to prepare the paper (e.g., the font style and size, citation style … etc.).

Answer: The paper has been modified and checked in accordance with the instructions of the journal.

Kind regards,

Round 2

Reviewer 1 Report

All questions answered and paper can be accepted